# A novel method (RIM-Deep) for enhancing imaging depth and resolution stability of deep cleared tissue in inverted confocal microscopy

Yisi Liu[1], Pu Wang[2], Junjie Zou[3], Hongwei Zhou[1,4]*

[1]Microbiome Medicine Center, Department of Laboratory Medicine, Zhujiang Hospital, Southern Medical University, Guangzhou, China; [2]Nikon Precision Corporation, Guangzhou , China, Shanghai, China; [3]Department of Neurosurgery, Zhujiang Hospital, Southern Medical University, The National Key Clinical Specialty, Guangzhou, China; [4]State Key Laboratory of Organ Failure Research, Southern Medical University, Guangzhou, China

## eLife Assessment

This study describes a **useful** technique to improve imaging depth using confocal microscopy for imaging large, cleared samples. The work is supported by **solid** findings and will be of broad interest to many microscopical researchers in different fields who want a cost effective way to image deep into samples.

*For correspondence:
biodegradation@gmail.com

Competing interest: The authors declare that no competing interests exist.

**Abstract** The increasing use of tissue clearing techniques underscores the urgent need for cost-effective and simplified deep imaging methods. While traditional inverted confocal microscopes excel in high-resolution imaging of tissue sections and cultured cells, they face limitations in deep imaging of cleared tissues due to refractive index mismatches between the immersion media of objectives and sample container. To overcome these challenges, the RIM-Deep was developed to significantly improve deep imaging capabilities without compromising the normal function of the confocal microscope. This system facilitates deep immunofluorescence imaging of the prefrontal cortex in cleared macaque tissue, extending imaging depth from 2 mm to 5 mm. Applied to an intact and cleared Thy1-EGFP mouse brain, the system allowed for clear axonal visualization at high imaging depth. Moreover, this advancement enables large-scale, deep 3D imaging of intact tissues. In principle, this concept can be extended to any imaging modality, including existing inverted wide-field, confocal, and two-photon microscopy. This would significantly upgrade traditional laboratory configurations and facilitate the study of connectomes in the brain and other tissues.

## Introduction

Tissue clearing techniques enable 3D imaging of whole tissues and organs (*Ueda et al., 2020*). Unlike traditional histology, these techniques reduce light scattering and absorption, enhancing depth imaging capabilities and achieving single-cell resolution in deep samples (*Susaki et al., 2014*). These approaches are now widely used in neurobiology, developmental biology, immunology, and oncology. To achieve deep imaging of fully cleared tissues, various optical sectioning techniques for

three-dimensional imaging have rapidly developed, including light sheet microscopy, two-photon microscopy, and confocal microscopy (*Ueda et al., 2020*).

For effective 3D imaging of cleared tissues, it is crucial to utilize microscopes equipped with objectives that offer both low to medium magnification and long working distances (*Jonkman et al., 2020*). Objectives with low numerical aperture (NA) are often used to achieve these long working distances, but they face significant challenges in maintaining high axial resolution for deep tissue imaging (*Birk et al., 2017*). The main issue arises from light scattering and reflection at the interfaces within the sample and between the sample and the lens, leading to image distortion and blurring. This phenomenon, known as spherical aberration, occurs when light rays entering the periphery of the lens focus at a different point on the optical axis compared to those entering near the center, leading to a significant degradation in fluorescence signal intensity and image resolution (*Diel et al., 2020*). Using dry objectives with high-refractive-index mounting media or cleared tissues can exacerbate this issue, as the sample itself contributes to the aberration (*Gibson and Lanni, 1992*). Therefore, matching the refractive index (RI) of the objective's immersion medium to that of the sample's immersion medium is essential to mitigate axial distortion caused by spherical aberration.

Many immersion objectives feature correction collars that can be adjusted to compensate for RI mismatches. Achieving an exact match between the refractive indices of the immersion media and the sample medium is often challenging. Although dipping caps with glass windows can transform air objectives into immersion lenses by enabling the front lens to be directly inserted into the sample media, this approach helps to avoid axial deviation during imaging. However, immersion objectives on inverted microscopes encounter limitations (*Dodt et al., 2015*). These objectives cannot be inserted into sample chambers and must move up and down outside the chamber, which restricts the imaging depth.

To overcome these limitations, several alternative methods have been developed. One such approach involves moving the sample within a chamber filled with mounting media, while the imaging objective remains fixed outside the chamber (*Engelbrecht and Stelzer, 2006*). Another technique utilizes adaptive optics, such as deformable mirrors or spatial light modulators, to correct aberrations across the field of view (*Ji, 2017*). Additionally, radially symmetric phase masks are used to balance spherical aberration at various depths within the sample, thereby simplifying the image restoration process through post-capture deconvolution (*Patwary et al., 2016*).

However, inverted confocal microscopes encounter issues due to the gravitational dispersion of the RI buffer between the objective and the confocal dish, resulting in RI mismatches with air and complicating high-resolution imaging. Additionally, the use of long-working-distance air objectives causes severe spherical aberrations due to RI mismatches, further limiting imaging depth and resolution.

Here, we developed the Refractive Index Matching-Deep (RIM-Deep) system to address these challenges, which includes an immersion chamber designed around the optical components of the objective lens and a specimen holder integrated with a motorized x–y–z stage. This design effectively stabilizes the refractive index between the objective and the sample media. When paired with an immersion objective, the RIM-Deep system enables high-resolution, deep imaging, significantly enhancing the capabilities of inverted confocal microscopy. This innovation provides a practical solution for achieving high-quality 3D imaging of cleared tissues, paving the way for more detailed and accurate biological studies.

## Results

Official adapters for 10 X immersion objectives in Nikon inverted confocal microscopy significantly enhance cleared tissue imaging, but RI buffer leakage still occurs, causing RI mismatch and limiting imaging depth (*Figure 1A*, *Figure 1—figure supplement 1A*). In response to this challenge, the RIM-Deep was designed based on a light sheet microscopy imaging chamber. It features a media reservoir affixed to the objective and filled with RI buffer, enabling 3D imaging of the sample within an imaging buffer. To further optimize imaging performance, a support bracket integrated into the motorized stages elevates the imaging platform and centrally positions a specimen holder (*Figure 1B*, *Figure 1—figure supplement 1B* and *Figure 1—figure supplement 2*).

To demonstrate the effectiveness of the RIM-Deep, imaging resolution was characterized by comparing 3 μm diameter fluorescence beads using standard adapter versus the RIM-Deep in 10 X immersion objective (*Figure 1A and B*). Within the 2 mm imaging limit, 10 X immersion objectives

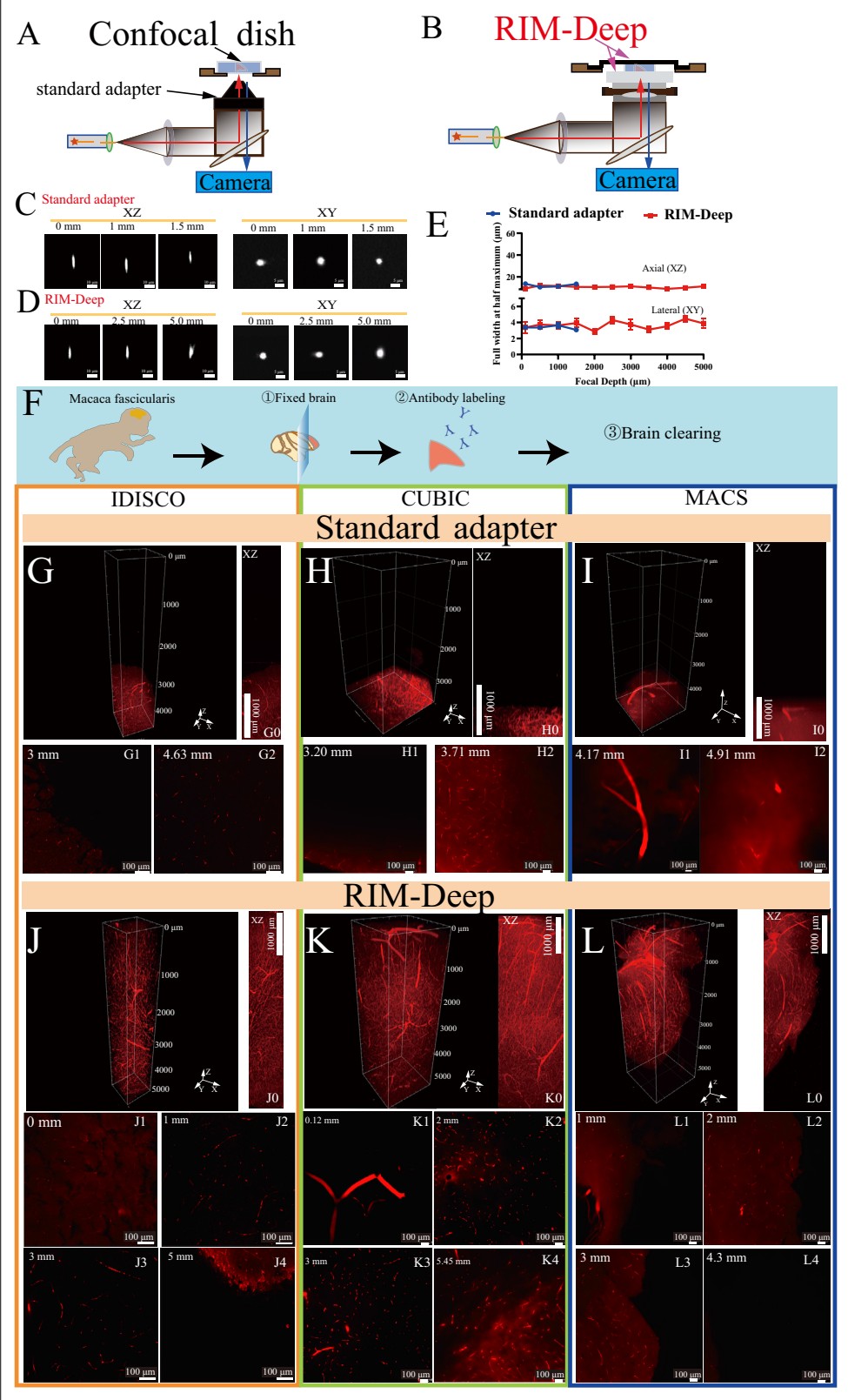

**Figure 1.** Resolution characterization and image depth of standard adapter and Refractive Index Matching-Deep (RIM-Deep). (**A and B**) Schematic diagram of a 10 X immersion objective with standard adapter (**A**) or RIM-Deep (**B**). (**C–D**) MIP of 3-μm-diameter beads imaged in the *xy* and *yz* planes using standard adapter (**C**) or RIM-Deep (**D**) at different axial positions. (**E**) Axial resolutions for a 10 X immersion objective paired with standard adapter or

*Figure 1 continued on next page*

*Figure 1 continued*

RIM-Deep at different axial positions. The resolution is estimated by FWHMs of intensity profiles with a Gaussian fit for 3-μm-diameter beads embedded in 1% agarose dissolved in CUBIC mounting solution. Data are presented as mean ± s.e.m. (n=10). (**F**) The experimental scheme for the brain clearing process in Macaca fascicularis. (**G, H, I**) Three-dimensional reconstructions of the Macaca fascicularis brain vasculature using three different tissue clearing methods (iDISCO, CUBIC or MACS) with standard adapter. (**G0, H0, I0**) MIP of G, H, or I in *xz* plane. (**G1– G2, H1–H2, I1–I2**) Optical section of (**G, H, I**) at varying depths. (**J, K, L**) Three-dimensional reconstructions of the Macaca fascicularis brain vasculature using iDISCO, CUBIC, or MACS with the RIM-Deep. (**J0, K0, L0**) MIP of J, K, or L in *xz* plane. (**J1–J4, K1–K4, L1–L4**) Optical section of (**J, K, L**) at varying depths.

The online version of this article includes the following video, source data, and figure supplement(s) for figure 1:

**Source data 1.** Source data for FWHM axial and lateral resolution comparison between RIM-Deep and Standard adapter across different focal depths.

**Figure supplement 1.** Illustration and actual picture of standard adapter (**A**) and the Refractive Index Matching-Deep (RIM-Deep) (**B**) mounted on the Nikon AXR microscope.

**Figure supplement 2.** Design and configuration of Refractive Index Matching-Deep (RIM-Deep) for inverted confocal microscope.

**Figure supplement 3.** High-depth imaging of Macaca fascicularis brain vasculature using Leica STELLARIS 5 with the Refractive Index Matching-Deep (RIM-Deep).

**Figure 1—video 1.** Three-dimensional reconstruction of the Macaca fascicularis brain vasculature using iDISCO with standard adapter.
https://elifesciences.org/articles/101143/figures#fig1video1

**Figure 1—video 2.** Three-dimensional reconstruction of the brain vasculature in Macaca fascicularis was achieved using CUBIC with standard adapter.
https://elifesciences.org/articles/101143/figures#fig1video2

**Figure 1—video 3.** Three-dimensional reconstruction of the brain vasculature in Macaca fascicularis using the MACS method with standard adapter.
https://elifesciences.org/articles/101143/figures#fig1video3

**Figure 1—video 4.** 3D reconstruction of the brain vasculature in Macaca fascicularis utilizing iDISCO with the Refractive Index Matching-Deep (RIM-Deep).
https://elifesciences.org/articles/101143/figures#fig1video4

**Figure 1—video 5.** Three-dimensional reconstruction of the Macaca fascicularis brain vasculature employing CUBIC with the Refractive Index Matching-Deep (RIM-Deep).
https://elifesciences.org/articles/101143/figures#fig1video5

**Figure 1—video 6.** Three-dimensional mapping of the Macaca fascicularis brain vasculature using MACS in combination with the Refractive Index Matching-Deep (RIM-Deep).
https://elifesciences.org/articles/101143/figures#fig1video6

**Figure 1—video 7.** High-depth imaging of Macaca fascicularis brain vasculature using Leica STELLARIS 5 with Refractive Index Matching-Deep (RIM-Deep).
https://elifesciences.org/articles/101143/figures#fig1video7

with the standard adapter maintain both optical axial (XZ) and lateral (XY) resolution (*Figure 1C and E*). As expected, the RIM-Deep also maintained optimal axial resolution across 5 mm (*Figure 1D and E*). However, there was no significant difference in lateral resolution between the two types of chambers (*Figure 1C–E*).

Subsequently, the high-depth imaging performance of the RIM-Deep was evaluated and compared to the standard adapter. Imaging was performed on a Macaca fascicularis brain section cleared using the iDISCO, CUBIC, and MACS methods (*Figure 1F*). Gravity affected the refractive index matching fluid at the tip of the objective, limiting the imaging depth to less than 2 mm (*Figure 1G–I2*, *Figure 1— videos 1–3*). As observed, with the RIM-Deep, the specimen holder was fully immersed in the RI liquid, the depth of images is obviously enhanced to about 5 mm, and the vessel structures could thereby be observed clearly (*Figure 1J–L4*, *Figure 1—video 4*, *Figure 1—video 5* and *Figure 1—video 6*).

The performance of the RIM-Deep method was also evaluated using a Leica inverted confocal microscope. Notably, high-resolution imaging of multiple cross-sections was successfully achieved

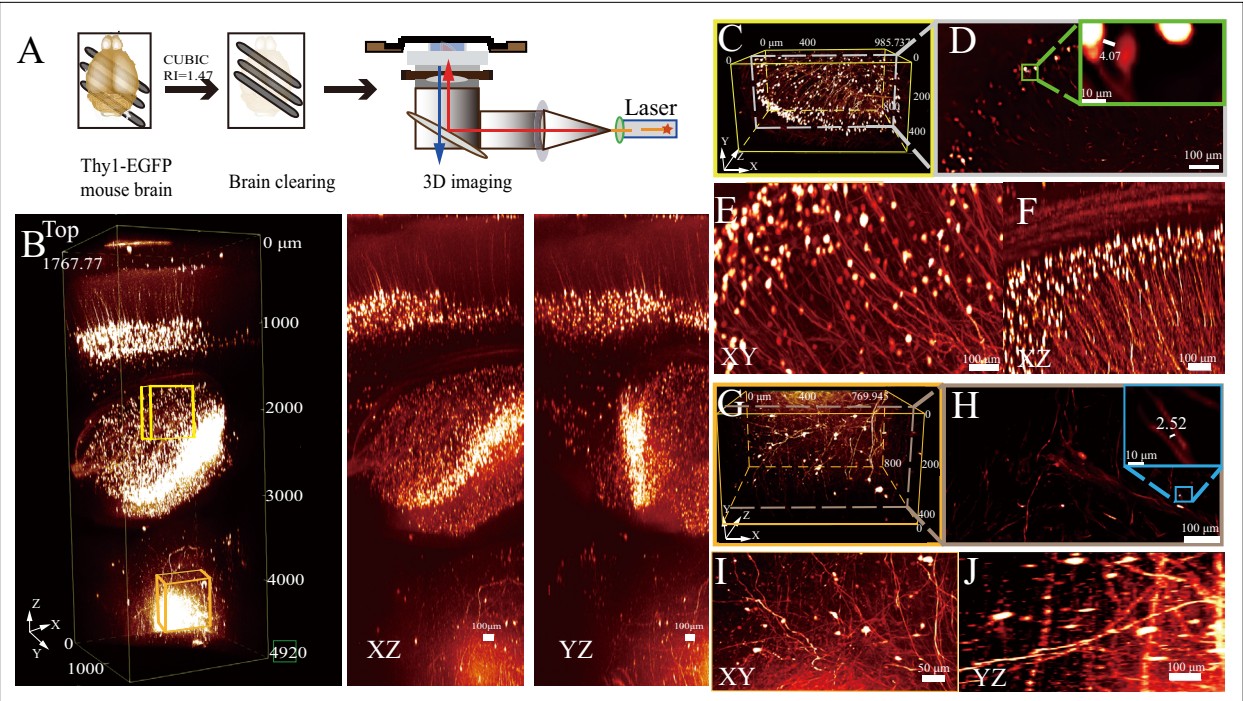

**Figure 2.** Deep imaging of a Thy 1-EGFP mouse brain using Refractive Index Matching-Deep (RIM-Deep). (**A**) Experimental scheme. (**B**) 3D reconstruction of the ~5 mm deep in the mouse brain (left), MIP views in XZ (middle) and YZ (right). (**C and G**) 3D reconstructions of neuronal structures within the hippocampus (C, yellow box) and thalamus (G, orange box), respectively, as indicated in B. (**D and H**) Lateral slices through the indicated lateral planes in (**C and G**). Zoom-in views of the selected areas in top right. (**E–F**) MIP view of hippocampus(C) in *xy* and *xz*. (**I–J**) MIP view of thalamus (**G**) in *xy* and *yz*.

The online version of this article includes the following video for figure 2:

**Figure 2—video 1.** Deep imaging of a single field in cleared Thy1-EGFP mouse brain tissue.

https://elifesciences.org/articles/101143/figures#fig2video1

for a Macaca fascicularis brain vasculature at depths of up to 5 mm, demonstrating consistent results across different imaging platforms (*Figure 1—figure supplement 3*, *Figure 1—video 7*).

To further assess the resolution and depth robustness of RIM-Deep in neuronal imaging, experiments were conducted on Thy1-EGFP mouse brains cleared with the CUBIC method (*Figure 2A*). The use of the RIM-Deep also significantly extended the imaging depth to approximately 5 mm (*Figure 2B*, *Figure 2—video 1*). This extension enabled the clear visualization and reconstruction of neuronal soma and axon structures in the hippocampus and thalamus (*Figure 2C, E, F, G, I*), maintaining resolution at the micron level (*Figure 2D and H*).

A 3×3 stitching mode with multi-layer Z-axis scanning was used to examine a cleared brain segment (*Figure 3A and B*, *Figure 3—video 1*). This method enabled visualization of neuronal cell bodies and axons up to nearly 5 mm in X-Y optical slices (*Figure 3C*). For large tissue imaging, the RIM-Deep was used on a cleared half-brain, achieving comprehensive imaging (*Figure 3D*). Its stable and precise positioning ensured accurate alignment across all optical slices, facilitating detailed and thorough imaging (*Figure 3E–I*).

Stroke, caused by disrupted brain blood supply, induces cell death and significant vascular changes (*Sekerdag et al., 2018*). Generally, confocal microscopy is limited to thin sections and cannot fully study vascular networks. However, with the assistance of the RIM-Deep, complete observation of vascular damage is possible. Therefore, mouse brains with induced cortical ischemia were examined to thoroughly investigate this process. Using VALID labeling and MACS tissue clearing techniques, the cerebral vasculature was reconstructed, revealing a significantly reduced vessel density in the ischemic cortex (*Figure 3J–L*, *Figure 3—video 2*, *Figure 3—video 3*, *Figure 3—video 4*). These findings demonstrate the effectiveness of the RIM-Deep in visualizing disease-related changes in vascular structures, making it a valuable tool in neuroscience research.

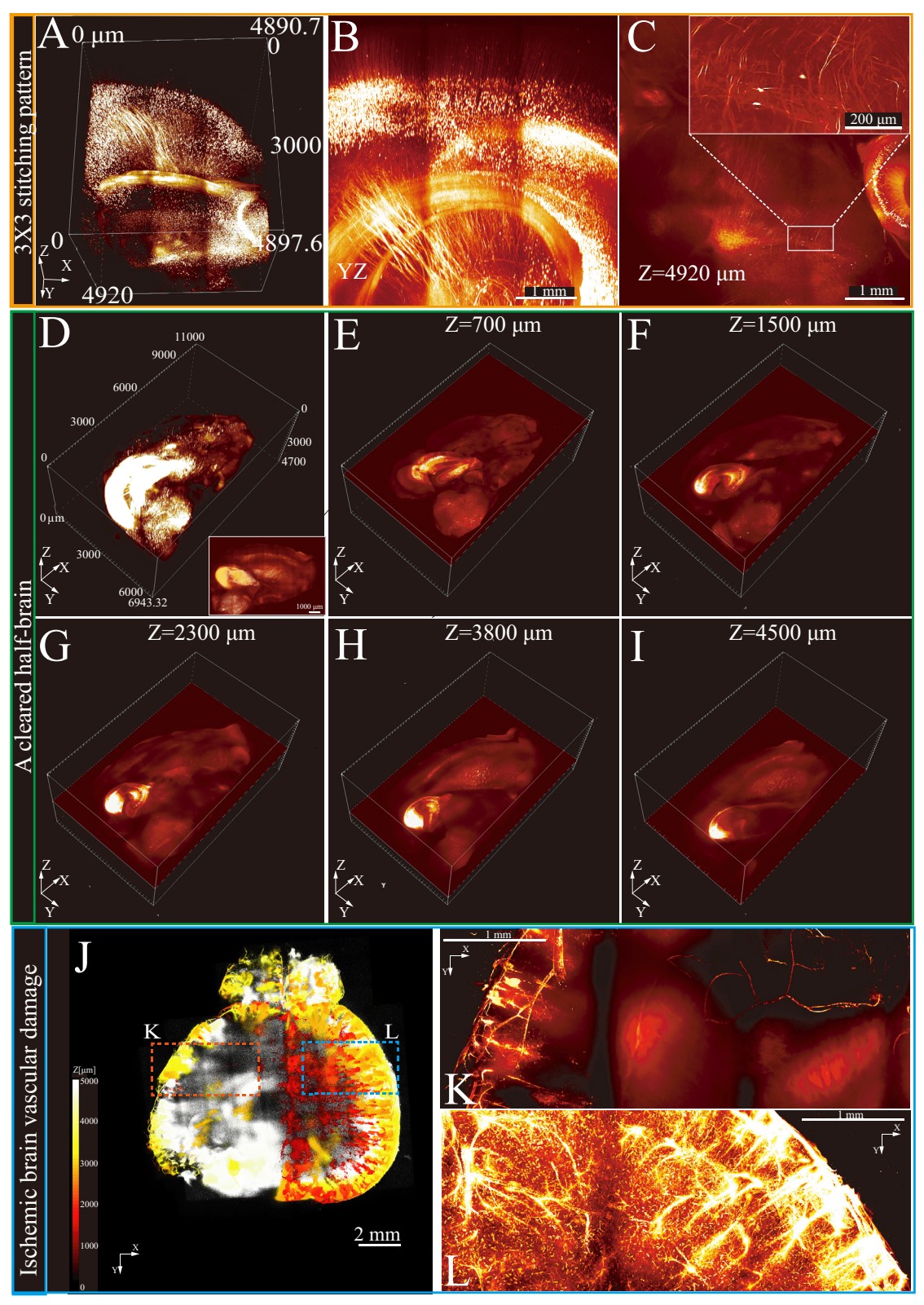

**Figure 3.** 3D imaging and reconstruction of neural and vascular structures in intact brain tissues using Refractive Index Matching-Deep (RIM-Deep). (**A**) 3×3 stitching pattern of deep imaging of a cleared brain. (**B**) MIP of *yz* side view of A. (**C**) Optical section of top layer in (**B**). Zoom-in views of the selected areas in top right. (**D**) 3D reconstruction of a half of cleared brain in a Thy1-eGFP mouse brain. The white box represents MIP. (**E–I**) Stitched

*Figure 3 continued on next page*

*Figure 3 continued*

single layer images in the Z-direction. (**J**) 3D imaging of the entire brain vasculature in ischemic stroke mice. The images along the z stack are colored by spectrum. (**K**) MIP of vascular imaging in the ischemic region (red box in **J**). (**L**) MIP of vascular imaging in the contralateral region (blue box in **J**).

The online version of this article includes the following video(s) for figure 3:

**Figure 3—video 1.** Deep imaging in 3×3 tiling mode of cleared Thy1-EGFP mouse brain tissue.

https://elifesciences.org/articles/101143/figures#fig3video1

**Figure 3—video 2.** Vascular network of the whole brain in MCAO mouse.

https://elifesciences.org/articles/101143/figures#fig3video2

**Figure 3—video 3.** Vascular network of the ischemic side in MCAO mouse brain.

https://elifesciences.org/articles/101143/figures#fig3video3

**Figure 3—video 4.** Vascular network of the control side in MCAO mouse brain.

https://elifesciences.org/articles/101143/figures#fig3video4

## Discussion

Inverted confocal microscopy often faces challenges such as limited axial resolution and spherical aberration when imaging cleared tissues. The RIM-Deep, featuring a media reservoir filled with RI buffer, was designed to address these issues. Experimental validations showed that the RIM deep maintained optimal axial resolution when imaging fluorescent beads. Using the three different tissue cleaning method on Macaca fascicularis brain tissues, imaging depths reached up to 5 mm with clear visualization of vascular structures. Similarly, in Thy1-eGFP mouse brains, the RIM deep extended imaging depth to approximately 5 mm, allowing clear visualization of neuronal and axonal structures. The RIM deep also effectively visualized vascular damage in mouse brains with cortical ischemia, proving valuable in neuroscience research. The RIM-Deep method also demonstrates versatility across different imaging platforms, ensuring consistent performance in various experimental setups.

The development of tissue clearing has revolutionized neuroscientific research. However, two-photon and light sheet microscopes are prohibitively expensive and their operation and maintenance costs are high, limiting their application scenarios. Many hospitals and laboratories, although capable of performing tissue clearing independently, are unable to conduct imaging. The inverted point-scanning confocal microscope, preferred for live cell and tissue slice imaging, offers advantages in deep imaging of cleared tissues over upright microscopes, including reduced sample contact and buffer use, lowering the risk of damage and cost. However, the use of long-working-distance air objectives introduces severe spherical aberrations due to RI mismatch, limiting imaging depth and resolution. Our RIM-Deep addresses these issues, extending imaging depth without compromising the microscope's normal functionality.

Wide-field microscopes, although efficient for rapid imaging, often face limitations in depth and resolution when applied to cleared tissues (*Shaw, 2006*). The RIM-Deep can also be employed in these cases to enhance imaging performance. When using wide-field microscopy with the RIM-Deep, it is necessary to apply deconvolution algorithms or other techniques to eliminate out-of-focus signals (*Shaw, 2006*). However, it is important to note that this system may reduce fluorescence intensity in deeper regions of the sample.

### Limitations of the study and prospect

Despite the advantages offered by the RIM-Deep, several limitations remain. The imaging speed of inverted point-scanning confocal microscopy is slower compared to light sheet microscopy, necessitating advanced post-processing algorithms for high-speed 3D imaging. To address this, resonance scanning methods combined with artificial intelligence-based image analysis can significantly improve the efficiency and accuracy of imaging, providing a more robust solution for detailed analysis (*Elliott, 2020*). Iterative reconstruction algorithms can enhance resolution and mitigate imaging artifacts. However, achieving uniform resolution across different depths remains challenging, particularly for very dense samples.

Furthermore, our research primarily focuses on neural tissues, and its effectiveness in imaging other tissue types remains to be explored. What's more, the objective lens used is limited to a correction collar for refractive indices between 1.33 and 1.51. Techniques like iDISCO and aqueous CUBIC/MACS, with refractive indices of 1.56 and 1.51/1.52, respectively, facilitate depth imaging to

nearly 5 mm. However, if the media's refractive index exceeds the lens's correction range, it could compromise resolution and image quality. Recently, laboratories have developed objectives with long working distances, high numerical apertures, and broad refractive index correction capabilities (*Voigt et al., 2024*). The Schmidt immersion objective, composed of spherical mirrors and aspheric correction plates, achieves an NA of 1.08 at a refractive index of 1.56, FOV of 1.1 mm, and W.D. of 11 mm (*Voigt et al., 2024*).

Future work should explore machine learning-based techniques to further optimize image quality and resolution (*Melanthota et al., 2022*). Moreover, as imaging depth increases, maintaining a high signal-to-noise ratio (SNR) becomes more challenging. Advanced denoising algorithms will be critical to address these issues, ensuring high-quality imaging of thick tissue samples (*Mandracchia et al., 2023*; *Li et al., 2023*).

In summary, the RIM-Deep represents a significant advancement in deep tissue imaging for inverted confocal microscopy, addressing key challenges related to axial resolution and spherical aberration. However, further refinements and integration with advanced computational techniques are necessary to optimize its performance and expand its applicability in various research fields.

## Materials and methods

### Animals

We used 6-y-old male Macaca fascicularis brain and male Thy1-EGFP mice brain gifted from Prof. Xiong Cao (South Medical University). Animals were housed in a temperature-controlled (24 ± 1°C) and humidity-controlled (50 ± 5%) facility with a 12 hr light/dark cycle (lights on at 8:00 a.m.) and had free access to standard chow and water. All animal procedures were approved by the Southern Medical University Animal Ethics Committee and the Ethics Committee of the Center of Guangdong Landao Biotechnology in Guangzhou, China (LDACU20170306-01) and was conducted in compliance with the guidelines of the Chinese Council on Animal Care to minimize animal suffering and reduce the number of animals used.

### CUBIC-based mouse brain clearing

We employed the CUBIC method for clearing Thy1-EGFP mouse fixed brain tissue (*Susaki et al., 2015*), with the process carried out at the Shared Instrument Platform of the School of Life Sciences at Tsinghua University.

### Mouse brain vascular labeling and tissue clearing using MACS

Adult male C57BL/6 J mice (8 wk) were used for the creation of the photothrombotic stroke model. Mice were anesthetized with sodium pentobarbital (50 mg/kg, IP). The head was secured in a stereotaxic device, and the skull was exposed. A photoactive dye, Rose Bengal was injected into the tail vein. Focused 561 nm laser irradiation was then applied to the cortical area to activate the dye, causing thrombosis and inducing ischemia by obstructing blood vessels. After surgery, mice were placed in warm cages for recovery and subsequently returned to their cages. Three days later, mice underwent deep sedation via intraperitoneal administration of sodium pentobarbital at a dosage of 50 mg/kg. Following this, vascular labeling was executed utilizing the VALID protocol (*Zhu et al., 2023*). Initially, mice were transcardiac infused with 0.01 M phosphate-buffered saline (PBS) for the removal of blood from the vascular system. Subsequently, a volume of 10–15 mL of VALID working solution, warmed to maintain its fluid state, was perfused through the circulatory system. Post perfusion, the mice were refrigerated at 4 °C overnight to solidify the gel within the vasculature. The next phase involved organ extraction and further post-fixation in 4% paraformaldehyde (PFA) for an extended period. Extreme care was taken to detach the skull from the perfused mouse body to avoid distortion of the sample. During this process, the dura mater was carefully excised. Finally, the removed skull was post-fixed in 4% PFA for an overnight duration.

MACS was executed for previously demonstrated (*Zhu et al., 2020*). Fixed samples were serially incubated in MACS-R0, MACS-R1, and MACS-R2 solutions, with gentle shaking at room temperature. MACS-R0 was prepared by mixing 20% (vol/vol) MACS with 15% (wt/vol) sorbitol (Sigma, 85529) in dH2O; MACS-R1 was prepared by mixing 40% (vol/vol) MXDA with 30% (wt/vol) sorbitol dissolved in 1×PBS; MACS-R2 was prepared by mixing 40% (vol/vol) MXDA with 50% (wt/vol) sorbitol in dH2O.

## Macaca fascicularis brain immunolabeling and clearing

As previously described (*Renier et al., 2014*), the Macaca fascicularis fixed brain underwent a series of washes and treatments as follows: Initially, they were washed twice for 1 hr each in PBS, followed by sequential hour-long immersions in 50%, 80%, and 100% methanol solutions, with the latter repeated. The samples were then bleached overnight at 4 °C using a 5% hydrogen peroxide solution in a 20% DMSO/methanol mix (prepared with 1 volume of 30% $H_2O_2$, one vol of DMSO, and four vol of methanol, all kept ice cold). Post-bleaching, the samples were washed twice in methanol for an hour each, followed by two 1 hr washes in a 20% DMSO/methanol solution, then in 80% and 50% methanol solutions for an hour each, and finally, twice in PBS for an hour. The last step involved two 1 hr washes in a PBS solution containing 0.2% Triton X-100, preparing the samples for subsequent staining procedures.

The samples underwent a series of incubation and washing processes as described: Initially, they were incubated overnight at 37 °C in a solution of PBS with 0.2% Triton X-100, 20% DMSO, and 0.3 M glycine. Following this, they were blocked using a PBS solution containing 0.2% Triton X-100, 10% DMSO, and 6% Donkey Serum at 37 °C for 3 d. Subsequently, the samples were washed twice for an hour each in a PBS solution with 0.2% Tween-20 and 10 µg/ml heparin (PTwH), then incubated in primary antibody (Anti-CD31, ab281583, RRID:AB_3096925) dilutions in a PTwH solution with 5% DMSO and 3% Donkey Serum at 37 °C for 4 d. After this incubation, the samples were washed in PTwH for 1 d, followed by an incubation in secondary antibody (Alexa Fluor 594, ab150080, RRID:AB_2650602) dilutions in a PTwH solution with 3% Donkey Serum at 37 °C for 4 d. The final step involved washing the samples in PTwH for 2 d before proceeding with clearing and imaging.

The immunolabeled tissues underwent a clearing process using the iDISCO technique, CUBIC or MACS. This method and the solutions employed were adapted from protocols detailed in a previously published study. The entire procedure was conducted at the Shared Instrument Platform located in the School of Life Sciences at Tsinghua University.

## Microscope and objectives

For image acquisition, we used the Nikon AXR inverted laser scanning confocal microscope and Leica STELLARIS 5 inverted laser scanning confocal microscope, paired with a Nikon 10 x/0.5 NA immersion objective.

## Design principle of the RIM-Deep

The RI adapter consists of a solution reservoir (*Figure 1—figure supplement 2A*), a specimen holder and support bracket (*Figure 1—figure supplement 2B*). The solution reservoir and specimen holder are made of glass. The nested arrangement of the solution and specimen holder forms a semi-closed space at the outer reservoir, which is filled with imaging buffer (*Figure 1—figure supplement 2C-F*). The specimen holder is centrally positioned on the top surface of the support bracket, ensuring that its base is parallel to the base of the support bracket. The bottom of the specimen holder is 0.17 mm thick cover glasses. The solution reservoir is nested in the cap of the objective.

## Image acquisition

Cleared tissue was placed in a 35 mm confocal dish or the dual-reservoir nested adapter. To ensure that the sample does not slide during X-Y scanning, the sample was fixed in place with transparent glass glue before imaging. Unless otherwise specified, the Z intensity correction module was used to offset the fluorescence signal degradation with depth. The NIS-Elements Advance research software was used for post-processing of images. The processed data were then imported into Imaris (Version 9.0.1, Bitplane AG) for the next step.

## Acknowledgements

This work was supported by grants from the National Key R&D Program of China (2022YFA0806400), Guangzhou Key Research Program on Brain Science (202206060001), and the National Natural Science Foundation of China (82130068) to H- WZ.

## Additional information

### Funding

| Funder | Grant reference number | Author |
|---|---|---|
| National Key Research and Development Program of China | 2022YFA0806400 | Hongwei Zhou |
| Guangzhou Key Research Program on Brain Science | 202206060001 | Hongwei Zhou |
| National Natural Science Foundation of China | 82130068 | Hongwei Zhou |

The funders had no role in study design, data collection and interpretation, or the decision to submit the work for publication.

### Author contributions

Yisi Liu, Conceptualization, Formal analysis, Visualization, Methodology, Writing - original draft, Writing – review and editing; Pu Wang, Conceptualization, Visualization, Methodology, Writing - original draft; Junjie Zou, Methodology, Writing – review and editing; Hongwei Zhou, Conceptualization, Supervision, Funding acquisition, Writing – review and editing

### Author ORCIDs

Yisi Liu ⓘ https://orcid.org/0000-0002-2121-8557
Hongwei Zhou ⓘ https://orcid.org/0000-0003-2472-8541

### Ethics

All animal procedures were approved by the Southern Medical University Animal Ethics Committee and conducted in compliance with the guidelines of the Chinese Council on Animal Care to minimize animal suffering and reduce the number of animals used.(LDACU20170306-01).

Reviewer #2 (Public review): https://doi.org/10.7554/eLife.101143.3.sa1
Author response https://doi.org/10.7554/eLife.101143.3.sa2

## Additional files

### Supplementary files

MDAR checklist

### Data availability

*Figure 1—source data 1* contains the numerical data used to generate the figures. All data generated or analysed during this study are included in the manuscript and videos; source data files have been provided for Figure 1 E.

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
