## [Editor Report · eLife Assessment]

This study describes a **useful** technique to improve imaging depth using confocal microscopy for imaging large, cleared samples. The work is supported by **solid** findings and will be of broad interest to many microscopical researchers in different fields who want a cost effective way to image deep into samples.

---

## [Referee Report · Reviewer #2 (Public review)]

Summary:

Liu et al investigated the performance of a novel imaging technique called RIM-Deep to enhance the imaging depth for cleared samples. Usually, the imaging depth using the classical confocal microscopy sample chamber is limited due to optical aberrations, resulting in loss of resolution and image quality. To overcome this limitation and increase depth, they generated a special imaging chamber, that is affixed to the objective and filled with a solution matching the refractive indices to reduce aberrations. Importantly, the study was conducted using a standard confocal microscope, that has not been modified apart from exchanging the standard sample chamber with the RIM-Deep sample holder. Upon analysing the imaging depth, the authors claim that the RIM-Deep method increased the depth from 2 mm to 5 mm. In summary, RIM-Deep has the potential to significantly enhance imaging quality of thick samples on a low budget, making in-depth measurements possible for a wide range of researchers that have access to an inverted confocal microscope.

Strengths:

The authors used different clearing methods to demonstrate the suitability of RIM-Deep for various sample preparation protocols with clearing solutions of different refractive indices. They clearly demonstrate that the RIM-Deep chamber is compatible with all 3 methods. Brain samples are characterized by complex networks of cells and are often hard to visualize. Despite the dense, complex structure of brain tissue, the RIM-Deep method generated high-quality images of all 3 samples given. As the authors already stated, increasing imaging depth often goes hand in hand with purchasing expensive new equipment, exchanging several microscopy parts or purchasing a new microscopy set-up. Innovations, such as the RIM-Deep chamber, hence, might pave the way for cost-effective imaging and expand the applicability of an inverted confocal microscope.

Weaknesses:

(1) However, since this study introduces a novel imaging technique, and therefore, aims to revolutionize the way of imaging large samples, additional control experiments would strengthen the data. From the 3 clearing protocol used (CUBIC, MACS and iDISCO), only the brain section from Macaca fascicularis cleared with iDISCO was imaged with the standard chamber and the RIM-Deep method. This comparison indeed shows that the imaging depth thereby increases more than 2-fold, which is a significant enhancement in terms of microscopy. However, it would have been important to evaluate and show the difference of the imaging depth also on the other two samples, since they were cleared with different protocols and, thus, treated with clearing solutions of different refractive indices compared to iDCISCO.

(2) The description of the figures and figure panels should be improved for a better understanding of the experiments performed and the thus resulting images/data.

(3) While the authors used a Nikon AX inverted laser scanning confocal microscope, the study would highly benefit from evaluating the performance of the RIM-Deep method using other inverted confocal microscopes or even wide-field microscopes.

Comments on Revision:

Regarding point (1)

Within the revised manuscript, Liu et al focussed on a more detailed comparison of the standard vs the RIM-Deep method of samples cleared with the 3 different methods.

Regarding point (2)

The revised description of the figures results in a better understanding of the data.

Regarding point (3)

The authors tested their method on different microscopic setups to show the compatibility.

Summary: the revised manuscript addressed all previously mentioned points.

---

## [Author Response]

The following is the authors’ response to the original reviews.

**Public Reviews:**

**Reviewer #1 (Public review):**
Summary:Liu et al., present an immersion objective adapter design called RIM-Deep, which can be utilized for enhancing axial resolution and reducing spherical aberrations during inverted confocal microscopy of thick cleared tissue.Strengths:RI mismatches present a significant challenge to deep tissue imaging, and developing a robust immersion method is valuable in preventing losses in resolution. Liu et al., present data showing that RIM-Deep is suitable for tissue cleared with two different clearing techniques, demonstrating the adaptability and versatility of the approach.

Greetings, we greatly appreciate your feedback. In truth, we have utilized three distinct clearing techniques, including iDISCO, CUBIC, and MACS, to substantiate the adaptability and multifunctionality of the RIM-Deep adapter.

Weaknesses:Liu et al., claim to have developed a useful technique for deep tissue imaging, but in its current form, the paper does not provide sufficient evidence that their technique performs better than existing ones.

We are in complete agreement with your recommendation, and the additional experiments will conduct a thorough comparison of the efficacy between the RIM-deep adapter and the official adapter in the context of fluorescence bead experiments, along with their performance in cubic and MASC tissue clearing techniques.

**Reviewer #1 (Recommendations for the authors):**
Suggestions for improvement:Major revisions:(1) For the bead experiment, the comparison was made to a 10X dry objective instead of an immersion objective, please make a comparison to the standard immersion objective.

Thank you for your suggestion. We fully agree with your suggestion to make a comparison with the standard immersion objective. We plan to conduct this comparison in future experiments and will thoroughly analyze the imaging differences between the official adapter and the RIM-deep adapter.

(2) It is unclear if an accurate comparison of objectives (same NA etc) is being made in Fig 1G-J, since the official adapter image appears to be of lower resolution even at the surface. At the very least, progressive 2D slices of the reconstruction must be shown for both adapters instead of just the RIM-Deep adapter.

Thank you for your suggestion. We strictly controlled the numerical aperture (NA) of the objectives in Fig 1G-J to ensure the accuracy of the comparison. However, the imaging resolution of the official adapter is consistent with that of the RIM-deep adapter. We agree that showing progressive 2D slices of the reconstruction would provide a more comprehensive comparison of the two adapters.

(3) Similarly, since there already exists an official adapter, it would be useful to see that RIM-Deep performs better even in the mouse tissue, since the clearing method was different.

Thank you for your suggestion. We will investigate the imaging performance of the two additional tissue clearing protocols using both the official adapter and the RIM-deep adapter.

(4) The movies need legends, as it is unclear if they even show 2-D slices very deep into the tissue.

Thank you for your suggestion. We will add figure legends to each movie.

(5) The purpose of Supplementary Figure 3 in its current form is unclear, as is the statement in the text related to it : "The effectiveness and utility of this adapter configuration have been substantiated through a comprehensive series of experimental validations".

Thank you for your suggestion. We will revise the statement to: "We validated the effectiveness and utility of this adapter configuration through a series of experiments."

(6) The system is variably referred to as RIM-Deep or DepthView Enhancer in the text and figures, it would be beneficial to the readers if the authors stuck to one name.

Thank you for your suggestion. We will choose RIM-Deep as the sole name.

Minor revisionsFigures(1) “Confocal" is incorrectly spelled as "confocol" in Figure 1, "media" is misspelled in multiple places.

Thank you. We will correct these errors.

(2) The camera is misplaced in the Figure 1 A drawing

Thank you. We will fix this issue.

(3) It would be useful to have actual pictures of the immersion objective setup (both RIM-Deep and the pre-existing adapter) since the diagrams are not very clear.

Thank you. We will include actual pictures of both the RIM-Deep and the pre-existing adapter in the supplementary materials.